# Topology-Informed Approaches to Enhanced Index Tracking

## Abstract

We address the problem of cross-market index tracking, replicating the performance of a foreign benchmark using only domestic assets, a task particularly relevant for markets with limited access to international investments. We propose a novel optimization framework that incorporates a topology-informed regularization term to extract persistent structural patterns from time-series price data. Our method leverages topological alignment between markets to construct robust index-mimicking portfolios without requiring constituent-level information. We further introduce a cost-aware formulation that accounts for transaction costs and their compounding effects. Empirical results on real-world data show notable gains over traditional tracking methods in both accuracy and robustness. Our approach holds broader potential for general time-series decomposition and synthesis.

## 1 Introduction

### 1.1 Financial Index Tracking Problem

Constructing efficient index-tracking portfolios, which aims to replicate the performance of a benchmark index without knowledge of its constituent assets, is a fundamental yet challenging problem in quantitative finance. Even when the constituent assets are known, full replication by holding all components of an index is often impractical due to substantial transaction costs, liquidity constraints, and portfolio management complexity. Furthermore, selecting an optimal subset of assets to minimize tracking error is inherently challenging, as it represents an NP-hard problem, rendering exact solutions computationally prohibitive in large asset universes.

Two distinct lines of research have emerged in addressing this problem. The first focuses on the development of heuristic or evolutionary algorithms to directly solve the constrained optimization formulations of index-tracking portfolios (e.g., Beasley et al., 2003; Chang et al., 2000; Canakgoz & Beasley, 2009). These methods offer practical near-optimal solutions and better scalability than exact methods, especially when facing large universes and complex constraints like cardinality and transaction costs. Second, another stream emphasizes matching the factor-level exposures of the index through factor models or related techniques (e.g., Lamont, 2001; Roll & Srivastava, 2018; Corielli & Marcellino, 2006b; Jiang & Perez, 2021). These approaches construct factor-mimicking portfolios by aligning portfolio exposures with the underlying risk factors of the benchmark, rather than its return, effectively replicating index performance with fewer assets and reduced variance (Fama & French, 1993; Roll & Srivastava, 2018).

There is a growing demand for novel methodologies that effectively address high dimensionality, redundancy, and the compounding effects of transaction costs in constructing robust index-tracking portfolios under limited information. Conventional approaches have yet to successfully tackle these challenges in a unified and principled manner. Recent studies have integrated machine learning and deep learning frameworks to enhance predictive accuracy and robustness in index-tracking problems (e.g., Shu et al., 2020; Dai & Li, 2024). Such approaches can incorporate alternative risk metrics and more general models, aiming to overcome the limitations of classical methods in quantitative finance. However, they still rely on various assumptions and may not fully capture the complex structure and dependencies within large-scale financial datasets. Importantly, replicating a non-standard financial index composed of non-tradable assets requires a fundamental understanding of its

underlying dynamics to achieve robust tracking performance, which standard methods consistently fail to accomplish.

## 1.2 TOPOLOGICAL DATA ANALYSIS FOR TIME-SERIES FEATURIZATION

A key innovation in our approach is the integration of Topological Data Analysis (TDA) to manage structural complexities inherent in the underlying dynamics of the target index. TDA, a rapidly developing area within data science and machine learning, leverages topological and geometric tools, particularly persistent homology, to identify meaningful structures in complex datasets. Recent studies have highlighted its unique strength in extracting salient features from highly dynamic time series data, where traditional correlation-based methods typically underperform (see, e.g., Gholizadeh & Zadrozny, 2018; Ravishanker & Chen, 2019; El-Yaagoubi et al., 2023; Chaudhari & Singh, 2023, for reviews). In the finance domain specifically, TDA has been effectively used in market regime detection, asset clustering, and systemic risk assessment, consistently demonstrating superior sensitivity and robustness to structural changes relative to conventional statistical methodologies (Gidea & Katz, 2017; 2018; Goel et al., 2020; Ruiz-Ortiz et al., 2022).

Despite growing interest in the application of TDA in financial analysis, its potential in the context of index tracking remains largely unexplored. A notable exception is the approach by Goel et al. (2020), which employs TDA as a preliminary filtering step to reduce the asset universe prior to applying conventional optimization techniques in a subsequent stage. However, direct integration of TDA tools into the index tracking problem has not been formally addressed in the literature, leaving a critical gap in methodologies for handling structural complexity in portfolio construction.

## 1.3 OUTLINE AND CONTRIBUTION

We specifically address a novel and challenging application scenario: tracking foreign market indices using only domestic stocks. This is particularly important for developing countries, where financial markets are relatively immature. Such cross-market indexing task is inherently complex as the domestic market may not fully span the risk factors underlying the foreign benchmark, requiring nuanced selection of proxy assets. Moreover, direct investment in the foreign market may be restricted or prohibitively expensive, further underscoring the importance of effective domestic replication strategies. Existing literature on the cross-market index replication primarily uses linear factor models or heuristic optimization methods, aiming to approximate foreign benchmarks using domestically available assets (e.g., Lamont, 2001; Roll & Srivastava, 2018; Errunza et al., 1999; Corielli & Marcellino, 2006a; Chavez-Bedoya & Birge, 2014; Hanschel et al., 2014). However, these methods predominantly rely on parametric factor models and often fail to account for structural differences and partial factor overlap between domestic and foreign markets, resulting in suboptimal tracking performance. Moreover, they fail to accurately account for the compounding effects of transaction costs in portfolio construction.

In contrast, we propose a comprehensive optimization-based framework that integrates topological data analysis (TDA) directly into the objective function as a regularization term. While traditional replication methods focus on minimizing local tracking errors, our TDA-based penalty captures persistent, macro-level structural patterns using features derived from persistent homology. Notably, our approach relies solely on price data to uncover latent topological correspondences between markets. By prioritizing domestic stocks that exhibit topological alignment with the movement of the foreign index, we construct robust *index-mimicking portfolios* capable of effectively tracking the target benchmark. Our key contributions are summarized as follows:

1. Novel TDA-based Methodology for Time-Series Analysis: We introduce a novel TDA-based framework for index tracking that captures complex interdependencies among assets by identifying persistent structural patterns in time-series dynamics. This enables the construction of robust index-mimicking portfolios, particularly effective in detecting transient correlations and regime shifts, features often missed by traditional methods. Importantly, beyond index tracking, our approach holds broader potential for general time-series decomposition and synthesis.

2. Cost-aware Index Replication: We propose a flexible optimization framework that explicitly incorporates transaction costs and their compounding effects into cross-market index replication, and demonstrate its effectiveness using real-world data. This significantly broadens the scope

of indexing strategies by extending conventional single-market approaches to more complex cross-market settings with critical trading constraints.

## 2 Preliminaries

### 2.1 Persistence Diagram and Landscape

In this section, we introduce the fundamental notations and key concepts from TDA that will be used throughout the paper. For further details and formal definitions, we refer the reader to Hatcher (2002); Edelsbrunner & Harer (2010); Chazal et al. (2016b) as well as Appendix A.

When inferring topological properties of a metric space $(\mathbb{X}, d)$ (usually a subset of a Euclidean space) from a finite collection $\mathcal{X}$ of observed points from it, we rely on the notion of simplicial complex, which can be seen as a high dimensional generalization of a graph. Given a set $V$, an *(abstract) simplicial complex* is a set $K$ of finite subsets of $V$ such that $\alpha \in K$ and $\beta \subset \alpha$ implies $\beta \in K$. Each set $\alpha \in K$ is called its *simplex*. The dimension of a simplex $\alpha$ is $\dim \alpha = \operatorname{card} \alpha - 1$, and the dimension of the simplicial complex is the maximum dimension of any of its simplices. Note that a simplicial complex of dimension 1 corresponds to a graph. Common constructions include the Vietoris–Rips (or Rips) complex and the Čech complex (see Definitions A.5 and A.6 in Appendix A.2 for formal definitions).

A *filtration* $\mathcal{F} = \{\mathcal{F}_a\}_{a \in \mathbb{R}}$ is a collection of subsets of $\mathbb{X}$ such that $a \leq b$ implies that $\mathcal{F}_a \subset \mathcal{F}_b$. Given a filtration $\mathcal{F}$ and for each $k \in \mathbb{N}_0 = \mathbb{N} \cup \{0\}$, we write $Dgm_k(\mathcal{F})$ to denote the persistence diagram corresponding to the $k$-th homological feature. Given two filtrations $\mathcal{F}$ and $\mathcal{G}$, the *bottleneck distance* between their associated persistence diagrams $Dgm_k(\mathcal{F})$ and $Dgm_k(\mathcal{G})$ is defined as

$$d_B(Dgm_k(\mathcal{F}), Dgm_k(\mathcal{G})) = \inf_{\gamma \in \Gamma} \sup_{p \in Dgm_k(\mathcal{F})} \|p - \gamma(p)\|_\infty,$$

where $\Gamma$ denotes the set of all bijections $\gamma : Dgm_k(\mathcal{F}) \cup Diag \to Dgm_k(\mathcal{G}) \cup Diag$, and $Diag$ represents the diagonal subset $\{(x, x) : x \in \mathbb{R}\} \subset \mathbb{R}^2$ with infinite multiplicity.

Persistence diagrams, as multisets of topological features, are challenging to analyze statistically due to their non-Euclidean structure, lack of differentiability, and sensitivity to noise. To overcome these limitations, recent work has introduced vectorized representations suitable for statistical and machine learning tasks. Among them, the *persistence landscape* (Bubenik, 2015; 2020) is a widely used, computationally efficient functional embedding into a separable Hilbert space.

**Persistence Landscape.** Given $Dgm_k(\mathcal{F})$, $\forall k$, we define a set of functions $l \in \mathbb{R} \mapsto \Lambda_p(l)$ for each birth-death pair $p = (b, d)$ in $Dgm_k(\mathcal{F})$ as follows:

$$\Lambda_p(l) = \max\{0, \min\{l - b, d - l\}\}. \tag{1}$$

For each birth-death pair $p$, $\Lambda_p(\cdot)$ is piecewise linear. Then the $j$-th order *persistence landscape* $\lambda_j : \mathbb{N} \times \mathbb{R} \to [0, L_{max}]$ of $Dgm_k(\mathcal{F})$ is defined as

$$\lambda_j(l) = \operatorname*{jmax}_{p \in Dgm_k(\mathcal{F})} \{\Lambda_p(l)\}, \quad j \in \mathbb{N}, l \in [0, L_{max}], \tag{2}$$

where jmax denotes the $j$-th largest value in the set.

### 2.2 State space reconstruction via time-delay embedding

We focus on time-series data, where the relevant topological structure is not directly observable in its raw form. A widely adopted and theoretically grounded approach to address this is *time-delay embedding*, motivated by Takens' embedding theorem (Takens, 1981). Originally developed for analyzing quasi-attractors in reconstructed state spaces, this method enables the recovery of underlying dynamical structures. For further details, we refer the reader to Garland et al. (2016).

As in many prior studies (e.g., Pereira & de Mello, 2015; Perea & Harer, 2015), we construct a time-delayed sliding window to extract geometric and topological features from temporal data. For simplicity, we focus on one-dimensional time-series observations. Let $f$ be a continuous function defined on the non-negative real line, $\mathbb{R}_{+0}$. Since featurization is performed over a finite time horizon,

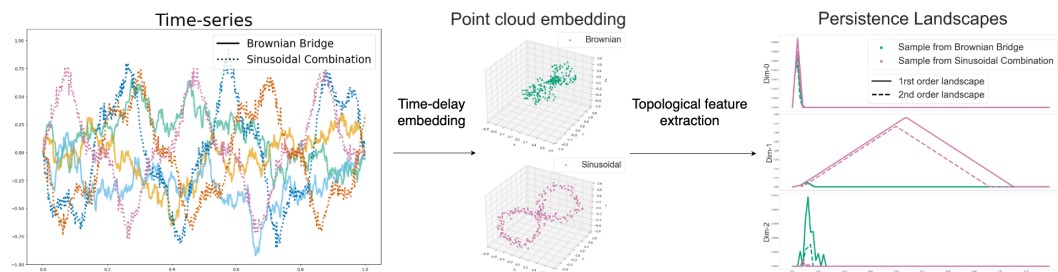

Figure 1: Time series samples generated from a Brownian bridge and a combination of sinusoidal functions (left). Corresponding point clouds obtained via time-delay embedding (middle). Persistence landscapes illustrating the distinct topological features of each process (right).

it suffices to consider $f(t)$ restricted to a compact interval $t \in [0, T]$, where $T \in (0, \infty)$. Then we define *sliding window operator* $SW_{m,\tau} f : \mathbb{R} \to \mathbb{R}^m$ by

$$SW_{m,\tau} f(t) \coloneqq \left[ f\left(t - (m-1)\tau\right), ..., f(t - \tau), f(t) \right]^\top.$$

In other words, for a fixed function $f$ the map $SW_{m,\tau} f$ generates the $m$ most recent samples up to time $t$ through an equi-interval sampling process with a predefined gap $\tau \in \mathbb{N}$. Thus, $\tau$ and $m$ can be interpreted as the delay parameter and the embedding dimension, respectively. Given $\tau$, $m$, and $N$, we construct the trajectory matrix $X_{m,\tau,N}^f \in \mathbb{R}^{\{N - (m-1)\tau + 1\} \times m}$ as follows:

$$X_{m,\tau,N}^f = \begin{bmatrix} SW_{m,\tau} f\left((m-1)\tau\right)^\top \\ SW_{m,\tau} f\left(1 + (m-1)\tau\right)^\top \\ \vdots \\ SW_{m,\tau} f\left(T\right)^\top \end{bmatrix} = \begin{bmatrix} x_0 & x_\tau & \cdots & x_{(m-1)\tau} \\ x_1 & x_{1+\tau} & \cdots & x_{1+(m-1)\tau} \\ \vdots & \vdots & \ddots & \vdots \\ x_{N-(m-1)\tau} & x_{N-(m-2)\tau} & \cdots & x_N, \end{bmatrix} \quad (3)$$

where we set $x_0 = f(0), \ldots, x_j = f(jT/N), \ldots, x_N = f(T)$. We also write $X_{m,\tau}^f$ if we only emphasize $m$ and $\tau$. The $m$-dimensional Euclidean space created by equation 3 is the sampled reconstructed state space induced by $f$, and the quasi-attractor of interest is then the topological pattern traced out by $X$ in $\mathbb{R}^m$. By Takens' embedding theorem, there always exists a pair $(\tau, m)$ such that the vectors generated via equation 3 are on a manifold topologically equivalent to the attractor of the original dynamical system for $f$ (Torku, 2016). Takens' embedding theorem allows one to assess the geometric structure of the attractor associated with an underlying dynamical system via the time-delay representation in equation 3. This reconstruction captures essential features of the quasi-attractor, which has proven particularly effective for characterizing signal periodicity in various applications (Robinson, 2014; Perea & Harer, 2015; Robinson, 2016). We refer interested readers to Appendix B for further details. The selection of delay-embedding parameters $(m, \tau)$ is discussed in Section 3.

### 2.3 MOTIVATING ILLUSTRATION

To motivate our approach, we illustrate how TDA effectively captures structural differences between time series generated by distinct stochastic processes: (i) a Brownian bridge, and (ii) a combination of sinusoidal functions with added low-amplitude noise. As shown in Figure 1, although quantitatively characterizing the differences between the two time series is nontrivial, their time-delay embeddings exhibit distinct topological structures that are clearly captured in the corresponding persistence landscapes, without the need for any learning procedures.

## 3 TOPOLOGICAL FEATURE EXTRACTION

In this section, we describe a procedure for extracting robust topological features from time-series data, which serve as informative representations of the intrinsic structure of the underlying process. Specifically, we obtain point cloud representations via time-delay embedding equation 3, and subsequently compute persistence landscapes equation 2 to capture topological features of the underlying dynamical system. This procedure can be summarized as the following three-step sequence.

---

**Algorithm 1:** Topological Time-series Feature Extraction

---

**Input:** Time series sequence $\{x_0, x_1, ..., x_N\}$

    1. Construct the point cloud $X \subset \mathbb{R}^m$ via the trajectory matrix equation 3 with $m, \tau$

    2. Compute the persistence diagrams $\{Dgm_k(X)\}_k$ (for either the Rips or the Čech filtration)

    3. From each $Dgm_k(X)$, compute persistence landscapes $\lambda_j$

**Output:** The vectors $\{\lambda_j(i\kappa)\}_{k,i}$ for $1 \leq k \leq K_{max}$, $1 \leq j \leq J_{max}$, and $0 \leq i \leq \lfloor L_{max}/\kappa \rfloor$.

---

Namely, the output is a vectorized representation of $\lambda_j(\cdot)$, discretely evaluated at resolution $\kappa$. Several variants of Algorithm 1 are also possible. For example, we may use alternative filtrations, such as Alpha complexes (for a formal definition, see Definition A.7 in Appendix A), and persistence landscapes can be replaced with other 1-Lipschitz vectorizations of persistence diagrams. The values of $K_{\max}$ and $J_{\max}$ are often user-specified, but in practice, they can be determined from data by increasing them until no additional significant patterns are observed. The performance of the algorithm is generally stable with respect to the choice of $L_{\max}$ and $\kappa$, as long as $L_{\max}$ is large enough to cover all persistence landscapes and $\kappa$ is sufficiently small to represent the shape of each landscape function.

To date, no consensus exists on selecting the embedding dimension $m$ and time delay $\tau$; existing heuristics remain nuanced and often subjective (Garland et al., 2016). A widely adopted approach for selecting $m$ is the false nearest neighbors (FNN) algorithm (Kennel et al., 1992), which we employ as the default method in this study, implemented via the tseriesChaos R package. Selecting an appropriate time delay $\tau$ is typically more subjective and often depends on expert judgment. Here, we adopt a hybrid of grid and random search for $\tau$ in a similar spirit to Kim et al. (2020). However, a range of established heuristics from the literature could also be used (see Garland et al. (2016); Bradley & Kantz (2015) for a comprehensive discussion). A detailed investigation is deferred to future work.

The computational bottleneck of Algorithm 1 lies in Step 3. For homology up to dimension $l - 1$, computing persistent homology using Rips or Čech complexes has time complexity $O(n^{2l} \log^3 n \log \log n)$. Using Alpha complexes reduces this to $O(n^{2\lceil l/2 \rceil} \log^3 n \log \log n)$ in the worst case, and to $O(n^2 \log^3 n \log \log n)$ under small random perturbations (Chen & Kerber, 2013; Boissonnat et al., 2018). Thus, Alpha complexes are often preferred due to their computational efficiency.

We now show how Algorithm 1 enables access to the topological features of the underlying signal process. Consider a true signal function $f : [0, T] \to \mathbb{R}$ from Section 2.2. We assume that the signal $f$ is corrupted by an additive signal noise $\zeta$, yielding the observed trajectory matrix $X_{m,\tau}^{f+\zeta}$. We are concerned with the topological features of the true, noise-free signal $f$. Thus, we analyze the robustness of the proposed topological featurization procedure to both noise and sampling, with the goal of recovering the intrinsic topological structure of $f$. The following lemma shows that the persistence diagram computed via Algorithm 1 remains close to the target persistence diagram $Dgm_k(f)$, and does not diverge arbitrarily under signal perturbations. Here, $Dgm_k(f)$ is the persistence diagram of the entire point cloud of the signal $\{SW_{m,\tau}f(t) : t \in [0, T]\}$.

**Lemma 3.1.** *Let $f : [0, T] \to \mathbb{R}$ be a Lipschitz function with Lipschitz constant $L_f$, and assume that $\|\zeta\|_\infty < \infty$. Then, we have that*

$$d_B(Dgm_k(X_{m,\tau}^{f+\zeta}), Dgm_k(f)) \leq \sqrt{m} \left( \|\zeta\|_\infty + \frac{L_f T}{N} \right), \qquad (4)$$

*where the persistence diagrams are computed using Rips or Čech filtration.*

The above result leads to the following stability result due to Bubenik & Dłotko (2017).

**Theorem 3.2.** *Let $\lambda_{j, X_{m,\tau}^{f+\zeta}}$ and $\lambda_{j,f}$ be the j-th order landscape functions from $Dgm_k(X_{m,\tau}^{f+\zeta})$ and $Dgm_k(f)$, respectively. Then,*

$$\|\lambda_{j, X_{m,\tau}^{f+\zeta}} - \lambda_{j,f}\|_\infty \leq \sqrt{m} \left( \|\zeta\|_\infty + \frac{L_f T}{N} \right). \qquad (5)$$

The bound in Theorem 3.2 arises from the discretization of observations in time and is inversely proportional to the number of observations $N$, with an additional contribution from the signal noise $\zeta$. Consequently, the result implies that for sufficiently large $N$ and small $\zeta$, the proposed method reliably approximates the topological structure of the underlying signal.

Lemma 3.1 and Theorem 3.2 rely on minimal assumptions, but they do not guarantee consistency due to the presence of $\|\zeta\|_\infty$. In the following, we impose an additional regularity condition, namely, that $\zeta$ is Lipschitz continuous, which ensures parameter stability and establishes the consistency of $Dgmk(X_{m,\tau,N}^{f+\zeta})$ as $N \to \infty$, and still much weaker than assuming zero noise, i.e., $\zeta = 0$.

**Lemma 3.3.** *Let $f, \zeta : [0, T] \to \mathbb{R}$ be a Lipschitz function with Lipschitz constant $L_f$ and $L_\zeta$, and suppose that the persistence diagrams are computed using Rips or Čech filtration. Then, we have that*

$$d_B(Dgm_k(X_{m,\tau,N_1}^{f+\zeta}), Dgm_k(X_{m,\tau,N_2}^{f+\zeta})) \leq \frac{\sqrt{m}(L_f + L_\zeta)T}{\min\{N_1, N_2\}},$$

$$d_B(Dgm_k(X_{m,\tau,N}^{f+\zeta}), Dgm_k(f + \zeta)) \leq \frac{\sqrt{m}(L_f + L_\zeta)T}{N}.$$

An analogous result holds for the landscape $\lambda_{j, X_{m,\tau,N}^{f+\zeta}}$ as well, as established in the following theorem.

**Theorem 3.4.** *Let $\lambda_{j, X_{m,\tau,N}^{f+\zeta}}$ and $\lambda_{j,f}$ be the $j$-th order landscape functions from $Dgm_k(X_{m,\tau,N}^{f+\zeta})$ and $Dgm_k(f)$, respectively. Then, we have that*

$$\|\lambda_{j, X_{m,\tau,N_1}^{f+\zeta}} - \lambda_{j, X_{m,\tau,N_2}^{f+\zeta}}\|_\infty \leq \frac{\sqrt{m}(L_f + L_\zeta)T}{\min\{N_1, N_2\}},$$

$$\|\lambda_{j, X_{m,\tau,N}^{f+\zeta}} - \lambda_{j,f+\zeta}\|_\infty \leq \frac{\sqrt{m}(L_f + L_\zeta)T}{N}.$$

Lemma 3.3 and Theorem 3.4 demonstrate that consistent robustness can be achieved: i.e., $\|\lambda_{j, X_{m,\tau,N}^{f+\zeta}} - \lambda_{j,f+\zeta}\|_\infty \to 0$ as $N \to \infty$ and $\|\lambda_{j, X_{m,\tau,N}^{f+\zeta}} - \lambda_{j,f+\zeta}\|_\infty \to 0$ as $N \to \infty$.

## 4 TOPOLOGY-INFORMED INDEX TRACKING

### 4.1 SETUP

We assume access to the price data of $N$ *basis assets* used to track the target index over $T + 1$ time points, $t = 0, 1, \ldots, T$. For each asset, $j = 1, \ldots, N$, the log returns are computed as $r_{jt} = \ln\left(\frac{P_{j,t}}{P_{j,t-1}}\right)$, $t = 1, \ldots, T$, where $P_{j,t}$ and $P_{j,t-1}$ denote the closing prices of the $j$-th asset on days $t$ and $t - 1$, respectively. We define the $T \times N$ matrix $R = [r_1, \ldots, r_N]$ to represent the return information of all basis assets, where each column vector $r_j = [r_{j1}, \ldots, r_{jT}]^\top$ denotes the return series of the $j$-th asset over the time horizon. We denote by $y \in \mathbb{R}^T$ the return series of the target index over the same time horizon. Starting from $t = 0$, our objective is to dynamically update, or *rebalance*, the portfolio weights $w = [w_1, \ldots, w_N]^\top$ assigned to the basis assets in order to closely track $y$, the performance of the target index.

**Loss Function.** Frequent rebalancing of portfolio weights may incur substantial transaction costs due to repeated buying and selling of assets. These costs can significantly impact the overall tracking error, particularly depending on the liquidity and trading characteristics of our basis assets. We let $f_b, f_s \in [0, 1]^N$ denote the vectors containing the fractional costs associated with buying and selling one unit of each asset, respectively. We incorporate transaction costs into our tracking procedure to account for their cumulative impact on portfolio performance over time, i.e., *compounding effects*. Let $C_t$ and $G_t$ denote the net asset value of the tracking portfolio and the total transaction cost incurred at time $t$, respectively. Given the portfolio weights prior to rebalancing, $w_{\text{prev}} \in [0, 1]^N$, and letting $b = \mathbb{1}(w > w_{\text{prev}}) \in \{0, 1\}^N$ denote the indicator vector identifying assets purchased during rebalancing, the transaction cost at time $t$ is given by $G_t = C_t\left[\{f_b \odot b - f_s \odot (1 - b)\}^\top (w - w_{prev})\right]$, where $\odot$ denotes the Hadamard (elementwise) product.

After rebalancing, the net asset value of the tracking portfolio is reduced by the transaction cost $G_t$. To offset this loss and ensure that the final net asset value over the prediction window aligns with

the hypothetical value that would have been achieved without transaction costs, one may consider deliberately overestimating the target index $y$ during tracking. Let $T_{\text{pred}}$ denote the length of the prediction window, i.e., the time horizon until the next portfolio rebalancing. The transaction cost compensation then can be formulated by introducing an adjustment factor $\alpha$ such that the net asset value at the end of the prediction window matches what would have been achieved in the absence of transaction costs. Specifically, we require

$$C_t \cdot \prod_{i=t}^{t+T_{pred}} \exp(y_i) = (C_t - G_t) \prod_{i=t}^{t+T_{pred}} \{\alpha \exp(y_i)\}$$

which yields the closed-form solution: $\alpha = (\frac{C_t}{C_t - G_t})^{1/T_{pred}}$. Consequently, the adjustment factor can be expressed as $\alpha = (\frac{C_t}{C_t - G_t})^{1/T_{pred}} = (\frac{1}{1 - \{f_b \odot b - f_s \odot (1-b)\}^\top (w - w_{prev})})^{1/T_{pred}}$, which is fully determined by the portfolio weights $w$, given that all relevant quantities $f_b$, $f_s$, and $w_{prev}$ are known, and the indicator vector $b$ is a deterministic function of $w$. Hence, we define the loss function as

$$\mathcal{L}(w) := \left\| y + \ln \left\{ \left( \frac{1}{1 - \{f_b \odot b - f_s \odot (1-b)\}^\top (w - w_{prev})} \right)^{1/T_{pred}} \right\} - Rw \right\|_2^2. \quad (6)$$

**Topological Regularization.** It has been shown that accurately tracking return movements based on recent historical data does not necessarily guarantee accurate tracking performance in the future (e.g., Corielli & Marcellino, 2006a; Roll & Srivastava, 2018). In contrast to previous approaches that rely on factor exposure matching, as discussed in Section 1.2, we address this issue by introducing a topological regularization framework. Specifically, building on the topological time-series featurization methods introduced in Section 3, we aim to minimize the topological discrepancy between the target index $y$ and the tracking portfolio $Rw$ by incorporating the following regularization term:

$$\mathcal{R}_{top}(y, Rw) = \sum_{k=0}^{K_{\max}} \sum_{j=1}^{J_{max}} ||\lambda_{j,y}^k - \lambda_{j,Rw}^k||_2^2, \quad (7)$$

where $\lambda_{j,y}^k$ and $\lambda_{j,Rw}^k$ denote the $j$th-order persistence landscapes of homological dimension $k$ computed from $y$ and $Rw$, respectively. While such topological regularization has proven effective for robust feature representation (Chen et al., 2019; Moor et al., 2020), it has not been formally explored in the context of time-series analysis, let alone in index tracking.

## 4.2 OPTIMIZATION

At each rebalancing step, we aim to solve the following optimization problem.:

$$\begin{aligned}
&\underset{w \in \mathbb{R}^N}{\text{minimize}} && \mathcal{L}(w) + \gamma \mathcal{R}_{top}(y, Rw) \\
&\text{subject to} && 0 \le w \le w_{\max}, \quad \mathbf{1}^\top w = 1, \\
& && b = \mathbb{1}(w > w_{\text{prev}}), \\
& && C_t \left[ \{f_b \odot b - f_s \odot (1-b)\}^\top (w - w_{prev}) \right] \le \delta.
\end{aligned} \quad (P)$$

Here, $\gamma \ge 0$ is a hyperparameter that governs the strength of the topological regularization term, while $w_{\max} \le 1$ specifies an upper bound on each individual asset weight; by default, we set $w_{\max} = 1$. The parameter $\delta$ is introduced to limit the total transaction cost incurred at each rebalancing step.

We now describe the gradient descent algorithm for solving equation P. Let $w^l$ denote the portfolio weights at iteration $l = 0, \ldots, n_{\text{iter}}$. The initial weights $w^0$ are set uniformly across basis assets with complete data, while assets with missing values, typically due to listing or delisting, are assigned $-\infty$. A softmax transformation is then applied to each $w^l$ to enforce the constraints $0 \le w_i \le w_{\max}$, and $\sum_{i=1}^N w_i = 1$. This transformation also effectively excludes assets with weights set to $-\infty$, as their softmax values become zero. Next, we compute the gradient $\nabla_{w^l} \left( \mathcal{L}(w^l) + \gamma \mathcal{R}_{top}(y, Rw^l) \right)$, which exists almost everywhere as shown in the following proposition.

**Proposition 4.1.** $\mathcal{L}(w) + \gamma \mathcal{R}_{top}(y, Rw)$ *from* P *is differentiable almost everywhere.*

---

**Algorithm 2:** Gradient Descent Algorithm for Cost-aware Topological Index Tracking

---

1   **Known parameters:** $f_b, f_s, w_{prev}$
2   **Hyperparameters:** $m, \tau, T_{pred}, K_{max}, J_{max}, \gamma, \rho$
3   **Input:** Log return of basis assets and target index: $R = [r_1, \ldots, r_N] \in \mathbb{R}^{T \times N}, y \in \mathbb{R}^T$
4   Initialize $w^0 \in \mathbb{R}^N$
5   **for** $l = 0, \ldots, n_{iter} - 1$ **do**
6      Normalize weights: $w^l \leftarrow \text{softmax}(w^l)$
7      Compute $\mathcal{R}_{top}(y, Rw^l; K_{max}, J_{max})$ from Algorithm 1
8      $l(w^l) := \mathcal{L}(w^l; f_b, f_s, w_{prev}, T_{pred}) + \gamma \mathcal{R}_{top}(y, Rw^l; K_{max}, J_{max})$
9      Gradient update with projection: $w^{l+1} \leftarrow \text{prox}_{\rho \imath_\mathcal{C}} \left( w^l - \rho \nabla_{w^l} l(w^l) \right)$
10   **Output:** Optimized weights: $w^{n_{iter}} \in \mathbb{R}^N$ .

---

We then apply the proximal gradient method to solve equation P. Define the set $\mathcal{C} := \{w \mid b = \mathbb{1}(w > w_{\text{prev}}), C_t \left[ \{f_b \odot b - f_s \odot (1 - b)\}^\top (w - w_{prev}) \right] \leq \delta \}$. This leads to the core update step of $w^{l+1} \leftarrow \text{prox}_{\rho \imath_\mathcal{C}} \left\{ w^l - \rho \nabla_{w^l} \left( \mathcal{L}(w^l) + \gamma \mathcal{R}_{top}(y, Rw^l) \right) \right\}$, where $\rho$ is the step size and $\text{prox}_{\rho \imath_\mathcal{C}}(\cdot)$ denotes the proximal mapping defined as $\text{prox}_{\rho \imath_\mathcal{C}}(z) = \arg\min_{x \in \mathbb{R}^N} (\imath_\mathcal{C}(x) + \|x - z\|_2^2 / 2\rho)$ , where $\imath_\mathcal{C}(x)$ is 0 if $x \in \mathcal{C}$ and $\infty$ if $x \notin \mathcal{C}$. The complete procedure is summarized in Algorithm 2, which is essentially gradient descent followed by projection onto $\mathcal{C}$. By the next proposition, $\mathcal{C}$ is indeed convex, ensuring that the projection onto $\mathcal{C}$ is uniquely and well-defined.

**Proposition 4.2.** $\mathcal{C}$ is a convex set.

## 5   CASE STUDY: TRACKING OFFSHORE MUTUAL FUNDS

We apply the proposed method to a novel cross-market index replication problem: tracking offshore mutual funds. Specifically, we aim to accurately replicate the performance of a U.S.-based mutual fund with undisclosed holdings using only exchange-traded funds (ETFs) that are tradable in the South Korean market. Despite its advanced economic status, South Korea's financial market is often classified at the level of an emerging or developing market (Mari, 2023). As a result, investors often seek exposure to foreign financial products; however, direct investment in offshore markets is frequently restricted by regulations or subject to prohibitively high costs. Consequently, there has been growing demand for indirect investment strategies, wherein investors seek to successfully replicate the performance of a target foreign index using highly liquid and low-cost domestic assets tradable in the Korean market.

We selected six U.S.-based actively managed funds that received a Gold rating from the Morningstar Medalist Rating system as of January 2024. As the basis asset pool, we consider all stock-based ETFs listed on the Korea Exchange. We use only price data, with no information about the target constituents, over the period from January 9, 2012 to June 24, 2024. The price data are aggregated at the weekly level, with each data point representing the weekly log return; however, we confirmed that using daily data yields consistent simulation results. For each rebalancing period, the model is trained on data from the preceding 104 weeks to predict outcomes over the subsequent 12 weeks. The training window is then shifted forward by 12 weeks, and portfolio weights are updated accordingly. This train–predict–rebalance procedure is iteratively applied until the entire dataset horizon is covered. In all cases, the fractional transaction cost for buying or selling one unit of each asset is fixed at 1%. Furthermore, we limit portfolio turnover to a maximum of 50% at each rebalancing point.

We compare the performance of six methods: (i) the factor-mimicking approach (FM); (ii) a return-based index-mimicking model without TDA regularization or transaction cost adjustment (RM); (iii) the linear programming–based index-tracking method of Canakgoz & Beasley (2009) (LM); (iv) the TDA-based asset-filtering approach of Goel et al. (2020) (TF); (v) a model incorporating TDA regularization only (TR); and (vi) a model with TDA regularization and compounded transaction cost adjustment (TT). (i) and (ii) correspond to the two conventional baselines discussed in Section 1.1. For (i), we employ the five-factor asset pricing model (Fama & French, 2015). The optimization-based models are trained with an initial learning rate of 0.5, which is reduced by a factor of 0.1 whenever the loss plateaus for 5 epochs. Training is terminated if no improvement is observed for

| Models | GSIHX | | | | PRWAX | | | |
|---|---|---|---|---|---|---|---|---|
| | MSE | Corr | $\beta_0$ | $\beta_1$ | MSE | Corr | $\beta_0$ | $\beta_1$ |
| FM | 0.030 | 0.830 | 1.489 ($\pm 0.377$) | 0.903 ($\pm 0.015$) | 0.021 | 0.895 | 1.636 ($\pm 0.389$) | 0.891 ($\pm 0.017$) |
| RM | 0.027 | 0.925 | 0.770 ($\pm 0.220$) | 0.990 ($\pm 0.009$) | 0.019 | 0.932 | 0.902 ($\pm 0.183$) | 0.988 ($\pm 0.008$) |
| LP | 0.026 | 0.908 | 0.748 ($\pm 0.211$) | 0.990 ($\pm 0.009$) | 0.019 | 0.925 | 0.879 ($\pm 0.177$) | 0.988 ($\pm 0.007$) |
| TF | 0.030 | 0.681 | 1.430 ($\pm 0.360$) | 0.910 ($\pm 0.016$) | 0.021 | 0.558 | 1.521 ($\pm 0.378$) | 0.897 ($\pm 0.017$) |
| TR(ours) | 0.026 | 0.970 | 0.742 ($\pm 0.218$) | 0.990 ($\pm 0.009$) | 0.019 | 0.968 | 0.873 ($\pm 0.182$) | **0.988** ($\pm 0.008$) |
| TT(ours) | **0.024** | **0.975** | **0.595** ($\pm 0.208$) | **0.991** ($\pm 0.009$) | **0.015** | **0.972** | **0.669** ($\pm 0.166$) | 0.986 ($\pm 0.007$) |

| Models | GOODX | | | | ARTKX | | | |
|---|---|---|---|---|---|---|---|---|
| | MSE | Corr | $\beta_0$ | $\beta_1$ | MSE | Corr | $\beta_0$ | $\beta_1$ |
| FM | 0.113 | 0.850 | 1.975 ($\pm 0.512$) | 1.679 ($\pm 0.029$) | 0.109 | 0.780 | 1.231 ($\pm 0.458$) | 0.914 ($\pm 0.013$) |
| RM | 0.098 | 0.904 | 1.012 ($\pm 0.423$) | 0.964 ($\pm 0.019$) | 0.089 | 0.773 | 0.745 ($\pm 0.387$) | 1.088 ($\pm 0.012$) |
| LP | 0.097 | 0.705 | 0.996 ($\pm 0.415$) | 0.965 ($\pm 0.018$) | 0.089 | 0.559 | 0.737 ($\pm 0.382$) | 1.087 ($\pm 0.012$) |
| TF | 0.110 | 0.635 | 1.820 ($\pm 0.500$) | 1.620 ($\pm 0.028$) | 0.108 | 0.585 | 1.221 ($\pm 0.436$) | 0.983 ($\pm 0.023$) |
| TR(ours) | 0.096 | **0.958** | 0.991 ($\pm 0.420$) | 0.967 ($\pm 0.019$) | 0.089 | **0.908** | 0.741 ($\pm 0.386$) | 1.087 ($\pm 0.012$) |
| TT(ours) | **0.091** | 0.942 | **0.904** ($\pm 0.409$) | **0.967** ($\pm 0.018$) | **0.088** | 0.901 | **0.625** ($\pm 0.384$) | **1.086** ($\pm 0.012$) |

| Models | TRIGX | | | | GQGPX | | | |
|---|---|---|---|---|---|---|---|---|
| | MSE | Corr | $\beta_0$ | $\beta_1$ | MSE | Corr | $\beta_0$ | $\beta_1$ |
| FM | 0.036 | 0.781 | 1.941 ($\pm 0.427$) | 0.951 ($\pm 0.021$) | 0.094 | 0.620 | 1.698 ($\pm 0.387$) | 0.962 ($\pm 0.017$) |
| RM | 0.027 | 0.890 | 1.415 ($\pm 0.215$) | 0.995 ($\pm 0.007$) | 0.083 | 0.728 | 1.159 ($\pm 0.389$) | 1.022 ($\pm 0.013$) |
| LP | 0.026 | 0.645 | 1.409 ($\pm 0.210$) | 0.996 ($\pm 0.007$) | 0.082 | 0.532 | 1.144 ($\pm 0.384$) | 1.021 ($\pm 0.012$) |
| TF | 0.035 | 0.585 | 1.822 ($\pm 0.431$) | 0.954 ($\pm 0.021$) | 0.093 | 0.502 | 1.689 ($\pm 0.385$) | 0.963 ($\pm 0.019$) |
| TR(ours) | 0.026 | **0.902** | 1.407 ($\pm 0.213$) | 0.997 ($\pm 0.007$) | 0.082 | 0.833 | 1.147 ($\pm 0.387$) | 1.021 ($\pm 0.012$) |
| TT(ours) | **0.024** | 0.884 | **1.207** ($\pm 0.204$) | **0.997** ($\pm 0.007$) | **0.081** | **0.855** | **1.041** ($\pm 0.385$) | **1.020** ($\pm 0.012$) |

Table 1: Performance on six U.S. mutual funds. 'Corr' reports the out-of-sample correlation between predicted and target returns. Better models yield lower MSE and higher Corr, with ideal calibration characterized by $\beta_0 \approx 0$ and $\beta_1 \approx 1$. Both MSE and $\beta_0$ are reported in $10^{-3}$ units.

more than 15 epochs, and hyperparameters of the topological regularization term are selected through a hybrid of grid and random search. We evaluate models using three criteria: (i) mean squared error (MSE) between predicted and target (log) returns; (ii) calibration via regressing predictions on targets, reporting the intercept $\beta_0$ and slope $\beta_1$ (optimal: $\beta_0 \approx 0$, $\beta_1 \approx 1$ (Canakgoz & Beasley, 2009)); and (iii) out-of-sample correlation between predicted and target returns.

**Results.** Table 1 shows that the proposed approaches achieve superior tracking performance. In particular, it appears that topological regularization consistently improves prediction accuracy and produces portfolios more closely aligned with the target index, as reflected in $\beta_0$ and $\beta_1$. By jointly exploiting local dynamics and the global topological structure of the time series, our methods enhance tracking accuracy, and transaction cost adjustments further improve performance by accounting for compounded trading losses. Figure 3 in Appendix C visualizes the tracking performance of our methods over the evaluation period.

## 6 DISCUSSION

We enhance conventional index-tracking methods by introducing topological regularization to capture persistent structures and accounting for compounding transaction costs, addressing key limitations of existing approaches. Variants of our strategy may also trigger rebalancing based on tracking error thresholds rather than fixed intervals. A key limitation is the computational cost associated with calculating persistence diagrams when $m$ is large. To address this, our forthcoming work proposes applying PCA to $X$ between Steps 1 and 2 in Algorithm 1, significantly reducing the computational burden. Additional future directions include improved hyperparameter tuning, alternative TDA descriptors (e.g., Euler curves), and broader real-world applications.

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
