# OpenReview forum: "Topology-Informed Approaches to Enhanced Index Tracking"
_ICLR.cc/2026/Conference — Submitted to ICLR 2026_

### Official Review · Reviewer_moBK · 2025-10-30

**Soundness:** 3
**Presentation:** 3
**Contribution:** 2
**Rating:** 4
**Confidence:** 4

**Summary:**

This paper proposes a topology-informed regularization framework that integrates topological data analysis into index tracking, with theoretical results on consistency and convergence and an available optimization algorithm. Experiments show promising performance in real-world data.

**Strengths:**

1. This study introduces topology-informed regularization to incorporate the concepts of topological data analysis into index tracking.
2. Theoretical analysis establishes the statistical consistency and convergence rate of the proposed optimization model.
3. An efficient optimization algorithm is further developed to solve the formulated problem, and experimental results demonstrate that the proposed model achieves superior performance.

**Weaknesses:**

1. Most of the theoretical developments presented in the paper have already been derived in existing literature.
2. The explanation of the comparative works in the experimental section is not sufficiently detailed.

**Questions:**

1. In lines 432 and following, the three tables refer to “LP” rather than “LM,” which was introduced earlier. Is this a typo or does “LP” denote a distinct concept? Please clarify the terminology consistency.
2. Line 426 describes the method as linear programming. However, according to the cited paper’s title, it may actually involve mixed-integer programming. Could the authors confirm which formulation is used?
3. Line 425 mentions the RM method but does not sufficiently explain how it works. Could the authors provide a clearer introduction or reference for this method?
4. Regarding line 236, the procedure for selecting parameters $\tau$ and $m$ is described as the default method implemented via the tseriesChaos R package. What is the practical guidance or implication of this choice for the experiment? The authors later say “$m$: fixed at 2 for simplicity” in the appendix—how do these statements align?
5. Continuing on this point: according to Takens’ embedding theorem, $m$ is related to the dimension of the underlying topological space. Simply fixing $m = 2$ (line 926) may not capture the correct structure. Would it be possible to add an experiment varying $m$ to test whether the results are robust or provide a brief discussion on whether $m = 2$ is sufficient?
6. Line 925 indicates a grid search over $\tau$ values {2, 5, 8}. Yet the main text (line 240) says “we adopt a hybrid of grid and random search for $\tau$ similar to Kim et al. (2020).” Where does the “random” component come in? The current experiment appears to be a simple grid search—could the authors clarify or expand on this search strategy?
7. The paper provides a theoretical proof of the statistical convergence rate of the proposed model. Could the authors include a synthetic experiment to empirically verify this convergence behavior?
8. In Appendix Figure 3, the experiments compare only the proposed variants against the target. Would it be possible to include the other related algorithms listed in Table 1 for a more comprehensive comparison?

---

### Official Review · Reviewer_kQAY · 2025-10-30

**Soundness:** 3
**Presentation:** 3
**Contribution:** 3
**Rating:** 8
**Confidence:** 4

**Summary:**

The paper tackles the challenge of cross-market index replication. The work introduces a topology-informed featurization which allows an effectiveregularization & optimization framework. The latter seems to well capture the structural dependencies within time-series data. This topology based approach is integrated into a cost-aware portfolio optimization model, balancing tracking accuracy, structural alignment and transaction costs. By prioritizing assets with topological similarity to the target index, the method achieves interesting results. The effectiveness of the method is shown over a series of experiments using real-world mutual fund data.

**Strengths:**

Innovative use of topological data analysis for structural time-series analysis and portfolio optimization.

Cost-aware formulation ensures practical relevance in high-friction markets. Good to have transaction costs and slippage as part of the optimization

Good theoretical underpinning of the work

Honest about limitations

**Weaknesses:**

Computational cost of persistence diagram computation may limit scalability.

Requires careful tuning of embedding parameters (\tau, m), which could be done better.

Interpretation of topological features in financial context is unclear.

Unrealism of some finance assumptions

No significance testing in results tables

**Questions:**

The “persistent structures” inferred from price data are not related to financial phenomena (e.g., sector rotation, volatility regimes).
Choice of \tau – you could avoid grid search by using methods like those proposed in A.C. Fowler, G. Kember. Delay recognition in chaotic time series. Physics Letters A, Volume 175, Issue 6, 1993

Whats the sensitivity to m, \tau etc?

Gradient descent – is there a more efficient natural gradient / coordinate descent method which avoids re-projections on to C at every step?

Your tables of results have no standard errors on MSE and Corr nor any attempt at significance analysis. Some of the [bold] results are unlikely to be significantly better than others. Can you add any measures of significance?

No runtime or complexity benchmark is reported.

The baselines are mostly legacy quantitative methods (factor models, LP optimization).

Have you considered comparisons to deep recurrent networks, transformers and so on, which have been shown very effective for time-series portfolio modeling. A baseline against any recent high-performance method would be useful.

The experiments use weekly data and limited ETFs, which may understate computational costs.

Homogeneous transaction costs (1%) and fixed turnover cap (50%) are overly simplistic.

The hyperparameter \gamma balancing the loss and topology terms is theoretically unmotivated. Can you offer any deeper insight?

Topological similarity between inferred attractors is likely related to similarity of rough path signatures. As the latter are widely used in creating cross-asset portfolios and asset baskets, have you considered the links?

---

### Official Review · Reviewer_iqg9 · 2025-11-02

**Soundness:** 3
**Presentation:** 2
**Contribution:** 2
**Rating:** 4
**Confidence:** 3

**Summary:**

This paper proposes a topology-informed framework for cross-market index tracking, constructing portfolios of domestic assets to replicate foreign benchmarks using only price data. The approach incorporates a TDA-based regularization term derived from persistence landscapes of time-delay embeddings into a cost-aware optimization that accounts for transaction costs and their compounding effects.

**Strengths:**

1. Innovative use of topological regularization, directly incorporating persistent homology into the tracking objective.

2. Provides stability results for the TDA features, lending theoretical support to the approach.

**Weaknesses:**

1. The motivation for incorporating persistent homology into index tracking is insufficiently justified. The paper does not clearly explain why topological features of price movements should improve tracking performance

2. Algorithm 2 relies on computing gradients and projections. But it is unclear how they are computed.

3. The TDA pipeline appears computationally heavy per iteration. The paper provides no empirical runtime analysis and comparison of computational costs against baselines.

4. Empirical scope is limited to six funds and one market-period setting, with no broader stress tests or ablations.

**Questions:**

Please refer to the Weaknesses above.

---

### Official Review · Reviewer_PBpv · 2025-11-02

**Soundness:** 3
**Presentation:** 2
**Contribution:** 2
**Rating:** 2
**Confidence:** 4

**Summary:**

This paper tackles the cross-market index tracking problem, where the goal is to replicate a foreign benchmark (e.g., U.S. index) using domestic assets (e.g., Korean stocks). The authors introduce a topology-informed optimization framework that incorporates Topological Data Analysis (TDA), specifically persistent homology and persistence landscapes, directly into the loss function as a regularization term. By enforcing topological alignment between the target index and the tracking portfolio, the model aims to capture persistent structural patterns in time-series dynamics that are often missed by correlation-based methods. Empirical results on real-world datasets demonstrate improved tracking accuracy and robustness compared to traditional heuristic and factor-based approaches.

**Strengths:**

1. Novel integration of TDA into optimization: Unlike prior work that used TDA merely for pre-filtering or feature extraction, this paper embeds TDA directly within the optimization loss, enabling end-to-end training that explicitly enforces structural alignment between the benchmark and the tracking portfolio.

2. Mathematical rigor: The theoretical development is solid. Lemma 3.1–3.4 establish stability and convergence properties, showing that the persistence landscape representation is robust to noise and sampling variation.

**Weaknesses:**

0. ICLR scope concern: Although methodologically interesting, the paper’s primary motivation and evaluation are within financial index tracking. The ML novelty lies mostly in applying known TDA concepts to a domain-specific optimization problem.

1. Limited baselines: The comparison is largely confined to traditional models (factor mimicking, LP-based optimization, or simple return matching). It omits more modern deep learning–based index tracking, such as Kim & Kim (2020, Quantitative Finance), and other time-series representation learning approaches (e.g., TS2VEC, Equity2Vec, SimStock). Without such comparisons, it is difficult to evaluate whether topology offers advantages over other nonlinear representation methods rather than just classical models.

2. Financial interpretation of topology is underdeveloped: While the paper provides a clear mathematical foundation for TDA, it lacks intuitive or financial justification for what topological features represent in economic or market terms (e.g., cyclicity, regime transitions, latent factor geometry).

**Questions:**

It’s unclear why the authors consider ICLR an appropriate venue for this paper. The references clearly show that nearly all prior work on index tracking (e.g., Beasley et al., 2003; Corielli & Marcellino, 2006; Kim & Kim, 2020) has been published in finance-oriented journals, not in general ML conferences. This suggests that the community evaluating such work would be finance or quantitative modeling researchers, rather than the ICLR audience, which typically expects fundamental ML innovation (e.g., new architectures, optimization theory, or representation learning frameworks).

---

### Official Review · Reviewer_YfGs · 2025-11-03

**Soundness:** 2
**Presentation:** 2
**Contribution:** 1
**Rating:** 2
**Confidence:** 3

**Summary:**

The authors tackle the problem of simulating index tracking. We propose a framework that uses a topology-informed regularisation term to extract persistent structural patterns from time-series price data.

To this end, the authors claim the following contributions:
1. Novel TDA-based Methodology for Time-Series Analysis
2. Cost-aware Index Replication

Of these, only the first is relevant to the ICLR community.

**Strengths:**

# originality
The application of the proposed TDA-based framework to time series is not entirely new; however, in the context of index tracking, some originality can be claimed.

# quality

The paper is well put together, tables, etc, are of sufficient quality

# clarity

The proposed method is clear.

# significance

The results are perhaps significant in the finance domain, but are not generally applicable to time series representation learning.

**Weaknesses:**

# Experiments are insufficient

To assess whether the proposed method is generally applicable, it would be beneficial to conduct experiments on data from other domains.

**Questions:**

How does this method work in general, such as when applied to other domains?

How much manual intervention and tweaking is required for the TDA?

---

### Meta-Review · Area_Chair_BkLj · 2026-01-06

**Summary:**

The technical contribution is limited -- the adaptation to index tracking does not need specific changes/improvements.

**Reviewer Concerns:**

N/A as no rebuttals were submitted.

**Reviewer Scores:**

N/A as no rebuttals were submitted.

---

### Decision · Program_Chairs · 2026-01-26

Reject